# Neuroscience Knowledge and Endorsement of Neuromyths among Educators: What Is the Scenario in Brazil?

**DOI:** 10.3390/brainsci12060734

**Published:** 2022-06-02

**Authors:** Estefania Simoes, Adriana Foz, Fernanda Petinati, Alcione Marques, Joao Sato, Guilherme Lepski, Analía Arévalo

**Affiliations:** 1Cancer Metabolism Research Group, Department of Cell and Developmental Biology, University of São Paulo, São Paulo 05508-000, Brazil; estefania.simoesfer@gmail.com; 2Department of Psychiatry and Medical Psychology, Federal University of São Paulo, São Paulo 04021-001, Brazil; fozadriana@gmail.com; 3Department of Psychotherapy, Institute of Psychiatry, University of São Paulo, São Paulo 05403-903, Brazil; petinaticoutinho@gmail.com; 4Department of Collective Health, Paulista School of Nursing, Federal University of São Paulo, São Paulo 04023-062, Brazil; alcionemq@gmail.com; 5Center for Mathematics, Computation and Cognition, Federal University of ABC, Santo André 09210-580, Brazil; jrsatobr@gmail.com; 6Department of Experimental Surgery, Medical School, University of São Paulo, São Paulo 01246-903, Brazil; lepski@gmail.com; 7Department of Neurosurgery, Eberhard Karls University, 72076 Tübingen, Germany

**Keywords:** neuroeducation, neuromyths, science education, pseudoscience, fake news, science literacy

## Abstract

The field of neuroscience has seen significant growth and interest in recent decades. While neuroscience knowledge can benefit laypeople as well as professionals in many different areas, it may be particularly relevant for educators. With the right information, educators can apply neuroscience-based teaching strategies as well as protect themselves and their students against pseudoscientific ideas and products based on them. Despite rapidly growing sources of available information and courses, studies show that educators in many countries have poor knowledge of brain science and tend to endorse education-related neuromyths. Poor English skills and fewer resources (personal, institutional and governmental) may be additional limitations in Latin America. In order to better understand the scenario in Latin America’s largest country, we created an anonymous online survey which was answered by 1634 individuals working in education from all five regions of Brazil. Respondents stated whether they agreed with each statement and reported their level of confidence for each answer. Significant differences in performance were observed across regions, between educators living in capital cities versus the outskirts, between those teaching in private versus public schools, and among educators teaching different levels (pre-school up to college/university). We also observed high endorsement of some key neuromyths, even among groups who performed better overall. To the best of our knowledge, this is the first study to conduct a detailed analysis of the profile of a large group of educators in Brazil. We discuss our findings in terms of efforts to better understand regional and global limitations and develop methods of addressing these most efficiently.

## 1. Introduction

Technological advances in recent decades have made neuroscience and its related fields one of the fastest growing areas of research. According to PubMed, an average of 3000 articles with the word “brain” were published per year in the mid-1960s [1], and in 2019, this number reached over 94,000. In the United States, 1990–2000 was called the Decade of the Brain, and although two decades have passed, interest in and the pursuit of knowledge in this field have not seemed to decrease. Nevertheless, believing in neuromyths, or false beliefs regarding the brain, seems to be as strong as ever [2,3].

While scientists worldwide rely on their academic training to analyze and critique information content (scientific or otherwise), laypeople are mostly left to navigate information online and judge whether new research, theories and discoveries are reliable. In a survey of prospective teachers in the southwestern United States, Zambo and Zambo [4] found that 64% of respondents reported using the internet as a source of information. As many as 66% of Brazilians have access to the internet (and 71% of those use at least one social media platform), compared to 59% (and 49%) of people globally. Brazilians are also above the global average in daily internet use (9.5 h/day vs. the 6 h and 42 min global average), with 85% of Brazilians accessing the internet daily. Furthermore, Brazil currently has over 31,000 communication companies [5] and approximately 16,500 online portals [6], which are mostly responsible for translating scientific research to non-technical language. Unfortunately, in Brazil, most journalists or other individuals who write about science do not necessarily have a science or medicine background.

Misinformation can also be attributed to the recent phenomenon of viral Fake News [7], which may lead entire communities to mistrust vaccines [8], or choose inappropriate medical treatments [9] or misguided educational methods [10]. In 2018, Brazil’s Ministry of Health launched a WhatsApp line to answer people’s questions regarding online information; their report revealed that 77% of the questions answered over one year were derived from fake news [11].

Neuroscience-related knowledge is relevant for a wide range of professions but is arguably critical for some specific lines of work, such as health and education. Knowledge about how the brain learns can improve education by helping teachers, professors and students adapt their teaching and learning strategies, respectively. With the right understanding of how the brain works, educators can positively influence individuals as well as help orient important educational policies. Importantly, neuroscience-related knowledge can protect educators from being deceived by pseudoscientific beliefs and products based on them, which often misguide students and can lead to the misuse of critical and often limited resources, financial and otherwise [12,13,14,15,16,17,18,19].

Several studies conducted in different parts of the world have assessed neuroscience knowledge among educators in those regions. Overall, findings have been surprisingly similar: most educators have limited neuroscience-related knowledge and relatively low confidence in that knowledge; furthermore, a large percentage endorse neuromyths [3,20,21,22,23,24,25,26,27]. Dekker et al. (2012) surveyed teachers in the UK and the Netherlands who reported being interested in the neuroscience of learning and found that educators believed in 49% of the neuromyths tested. Additionally, teachers most interested in neuroscience and who showed greater overall knowledge were actually more likely to endorse neuromyths. This finding may seem paradoxical at first, but a careful analysis suggests a possible explanation: a desire to know more may lead someone to seek more information, but when the information accessed is limited or incorrect (and the person does not have the skills required to judge it), they may be more likely to endorse pseudoscientific claims. Thus, could seeking knowledge without the proper tools to evaluate it be worse than not seeking it at all? In a world where fake information abounds, this may indeed be the case [3]. Herculano-Houzel’s survey about neuroscience literacy conducted with laypeople in Rio de Janeiro in 2002 also found that respondents who reported reading more did not necessarily obtain higher scores [28].

In a Spanish language adaptation of Dekker et al.’s survey, Gleichgerrcht et al. surveyed 3451 educators in seven Latin American countries [24]. As in Dekker et al., educators with greater overall knowledge also were more likely to endorse neuromyths. While those authors included a large sample of Latin American educators, they purposely excluded Brazilian educators, since their native language is Portuguese, and not Spanish. Thus, to date, we do not know how Brazilian educators fare on these matters. Sousa and Alves (2017) recently argued that most of the content and methods used in the training of Brazilian educators is outdated and in need of urgent reform [29].

Between 2015 and 2017, author AF was invited to talk about how the brain learns at 15 conferences and events geared towards educators that were held in all five regions of Brazil. There, she answered questions and spoke to teachers to gather information on what kinds of knowledge they seek as well as where the greatest doubts or misconceptions lie. In the northeastern state of Ceará, one teacher claimed the brain is used only for math and writing, and that children with bigger heads (presumably filled with water) are not able to learn as well. In Manaus, in the northern state of Amazonas, one math teacher claimed that teaching math has nothing to do with the brain. Yet, another teacher from the midwestern state of Tocantins claimed that reason and emotion are processed in the left and right hemispheres of the brain, respectively. Finally, a teacher from the North claimed emotion has to do with the heart and not the brain. AF compiled these questions into a survey, which she distributed at several conferences prior to her talks. The survey created for the current study was based on (1) that original survey, (2) similar surveys conducted in other countries and cited above, and (3) a previous study conducted by authors ES, FP, AA and GL with laypeople in Brazil that used Google tools to identify the terms most often searched for online by Brazilians in Portuguese [30].

Brazil is a country of continental proportions, in terms of size, number of different cultures, socioeconomic levels and access to resources [31]. The five regions in Brazil are unequally favored in terms of wealth and education: total years of schooling are significantly higher and illiteracy rates are significantly lower in the south and southeast relative to the north and northeast [31]. Currently, illiteracy rates are 3.3% in the south and southeast, 4.9% in the center-west, 7.6% in the north and 13.9% in the northeast [32]. A study conducted by the Brazilian Institute for Geography and Statistics (Instituto Brasileiro de Geografia e Estatística, IBGE) in 2018 revealed that internet use also varies across regions, with people in the southeast and center-west using internet the most (81.1% and 81.5%, respectively), followed by the southeast (78.2%), north (64.7%) and northeast (64%), and urban areas accessing the internet at a rate of 79.4% (versus 46.5% in rural areas). In terms of age, internet use is highest among 18–29-year-olds (90–91%) and lowest among individuals 60 and older (38.7%), with steadily declining numbers as age increases [33].

For the current study, we created an online survey containing 28 statements about the brain and learning. We shared the survey among colleagues and other contacts working in education and distributed it via several online platforms. A total of 1643 individuals who reported working in the area of education in Brazil provided anonymous answers to the survey, indicating either ‘agree’ or ‘disagree’ to each statement as well as the degree of confidence in their answers on a scale of 1–5. In line with other authors [34], we believed the confidence score would provide an additional piece of information that binary answers (e.g., true/false or agree/disagree) would not provide.

Our main goal with this survey was to obtain a clearer picture of neuroscience-related knowledge among Brazilian educators from different regions, types of school and teaching levels, in order to identify specific knowledge gaps as well as which neuromyths are endorsed the most. The information obtained would give us a picture of how Brazilian educators fare relative to other educators around the world and could also help guide efforts to improve scientific communication in the region. Furthermore, it could inform efforts to develop better-quality training programs and courses (e.g., undergraduate, graduate, extension, and free) designed specifically for educators. To the best of our knowledge, this is the first study evaluating neuroscience-related knowledge among a large group of educators in Brazil.

## 2. Material and Methods

### 2.1. Participants

A total of 1651 individuals who reported working in education provided online anonymous answers to the entire survey and provided information regarding age (20–71+), gender, region within Brazil (South, Southeast, Midwest, North, or Northeast), whether they lived in a capital city or in the outskirts, type of institution, private versus public, institutional role(s), teaching levels(s) (pre-school to college/university), and years in education (Figure 1A–H). Seventeen (*n* = 17) respondents indicated that they did not want their anonymous answers to be included in the research study and were thus excluded from the analyses. Therefore, all analyses were conducted on the responses provided by 1634 participants. Figure 1A–H show population distributions by group. The sample we obtained was strikingly representative of the population across Brazilian regions (71% Southeast, 10% Northeast, 13% South, 2% North and 4% Midwest) [31].

### 2.2. Instrument

The 28 statements included in the survey were selected based on a few different considerations and by tapping into a few different sources. First, we considered the classic neuromyths tested in several previous studies conducted in different parts of the world (e.g., [3,22]). Secondly, we used a survey created through knowledge gathered at several national conferences at which author AF presented between 2015 and 2017 (see Introduction). Finally, the third source used to create the survey was a previous study from our group that included 30 true/false statements about neuroscience and targeted laypeople from different professional areas [30]. In that study, each author created a list of common terms or keywords in the following areas most commonly covered in introductory neuroscience courses: anatomy, neurotransmitters, pathologies and disorders, exams, curiosities and myths, names of authors, drugs/medications and therapies, which yielded a total of 336 words. We then inserted those words into Keyword Planner within Google Ads (Google’s tool for creating ads on Google’s platform and networks) to identify the number of searches and clicks for those words in Brazil for an entire year (2018–2019). Next, we used the keywords with the largest click volume to conduct simple Google searches to identify the questions most often associated with those keywords within searches. In other words, through these most-clicked words, we were able to identify the most often searched phrases or questions, which inspired the creation of the statements in that survey. Because our main goal here was to gain insight into the state of knowledge among educators in Brazil (across regions, types of school and teaching levels, among other variables evaluated), we made sure to include the questions that evoke the most interest and doubts among Brazilians, as well as those previously tested elsewhere.

Once created, we divided the statements into the following seven categories: (1) brain characteristics, (2) executive and cognitive functions, (3) neurophysiology and learning, (4) emotion and learning, (5) literacy, reading and writing, (6) learning disorders, and (7) learning strategies and methods.

### 2.3. Procedures

The survey was created in Google Surveys and was distributed and made available online between 27 February and 13 May 2020, with a special effort to reach educational platforms and groups in all 5 regions of Brazil (Appendix A).

Order of presentation was balanced to avoid clusters of true or false answers or similar themes, and all participants viewed all 28 statements in the same order. Only after answering each statement could participants view and answer the following question. Participants were also asked to rate their confidence in their answer to each statement, on a scale of 1–5. Table 1 lists all 28 statements, overall response accuracy for each (correct and incorrect) as well as average reported confidence. Scores were converted to continuous variables (percent correct, 0–100%) across all 28 items for each participant and across all 1634 participants for each survey item. To facilitate visualization, we ranked the statements within the table from lowest to highest overall score.

The study was carried out in accordance with the Declaration of Helsinki. While ethical compliance varies across countries and institutions, online questionnaires to unidentified adults generally do not require IRB approval, which was the case at our institutions. In line with the Ethical Standards of the American Educational Research Association [35], the recommendations for good practice in designing internet-based research [36], and Mixed Methods Research Methodologies [37], for our online survey, we were transparent in recruiting, considered participant privacy and ensured secure communication protocols, obtained informed consent, allowed participants the opportunity to withdraw from the research at any time, and did not subsequently use the data for other practices. We also explained the study’s purpose, indicated that anonymity would be protected at all times by never collecting (or storing) names or any other identifying information and coding answers so that these could not be associated with a particular participant. The first page of the survey explained these issues and asked participants whether they agreed with their anonymous answers being used in the research study. As mentioned above, 17 participants stated that they did not agree to have their answers used and were thus excluded from all data analyses.

### 2.4. Data Analysis

All analyses were conducted using JMP 14.0 (SAS Institute, Cary, NC, USA). First, we analyzed the score distribution and tested for normality using the Anderson–Darling test. Since scores obeyed normal distribution, we performed multiple regression analyses as well as Analyses of Variance (ANOVA) to compare all groups, and pairwise Tukey–Kramer HSD tests for additional post hoc comparisons. All results are reported as mean ± standard error, test statistics as A2 (for Anderson–Darling), F (for ANOVA) or *p* values (for Tukey–Kramer HSD’s tests), and d (Cohen’s d) and η^2^ (partial eta squared) for effects sizes for significant ANOVAs conducted with two and more groups, respectively. Significance was set at *p* < 0.05.

## 3. Results

### 3.1. Quantitative Analyses

We first tested and confirmed the normality of the score distribution (Anderson–Darling test, A2 = 26.372, *p* < 0.0001; curve coefficients µ = 86.945 ± 0.168, σ = 6.795 ± 0.0460). Next, we wanted to know the effect of each of the variables of interest on participants’ performance. A multiple regression analysis revealed that the variables that contributed the most to participants’ performance were region (*p* = 0.0014), followed by time in education (*p* = 0.0052), then capital versus outskirts (*p* = 0.0147), and teaching level (*p* = 0.0150). Next, we present each of the main effects.

### 3.2. Main Effects

#### 3.2.1. Region

There was a significant association between region and participants’ scores, F(4,1629) = 5.05, *p* = 0.0005, η^2^ = 0.01, with individuals from the Southeast and South responding best (see Figure 2C and Appendix A). Additionally, post hoc Tukey tests revealed that individuals from the Southeast performed significantly better (87.4%) than those in the Midwest (84.8%; *p* = 0.0180) and Northeast (85.6%; *p* = 0.0140) (Figure 2C).

Moreover, individuals from capital cities performed significantly better than those living and teaching in the outskirts of major cities (F(1,1632) = 8.6, *p* = 0.0034, d = 0.15) (Figure 2D).

#### 3.2.2. Time in Education

Years of teaching experience was also associated with performance (F(7,1626) = 2.54, *p* = 0.0130, η^2^ = 0.01; Figure 3E; Appendix A), with the group that reported having more than 40 years teaching experience scoring highest (88.6%). No post hoc Tukey tests reached significance.

#### 3.2.3. Type of Institution

A main effect of type of institution was also significant, F(8,1625) = 2.06, *p* = 0.0360, η^2^ = 0.01. Individuals performing best were those working in federal public schools and private schools (88.8% and 87.4%, respectively; Figure 3A), and those performing worst worked in municipal public schools (86.4%) and state public schools (86.5%) (Appendix A). No post hoc Tukey tests reached significance for type of institution.

When we then grouped all types of institutions into public, private, or both, a one-way ANOVA test revealed significant differences among groups F(2,1631) = 2.97, *p* = 0.0500, d = 0.35 (Figure 3B; Appendix A), with individuals who declared working in private schools scoring significantly better than those working in public schools (post hoc Tukey test, *p* = 0.0390).

#### 3.2.4. Teaching Levels

There was also an association between grade level in which respondents taught and performance (F(9,1624) = 3.23, *p* = 0.0007, η^2^ = 0.02) (Figure 3D). Those teaching higher levels (college/university) scored 88.7%, while those teaching pre-school/kindergarten scored 85.7%, on average (see Appendix A). Additionally, post hoc Tukey tests revealed that individuals who declared teaching higher levels and multiple grades up to college/university obtained a significantly higher average score than those teaching pre-school/kindergarten (*p* = 0.0006 and *p* = 0.0057, respectively).

#### 3.2.5. Age, Gender and Institutional Roles

Age did not have a significant effect on participants’ performance (F(5,1628) = 0.96, *p* = 0.4401, see Figure 2A and Appendix A). There was also no effect for gender, F(2,1628) = 2.38, *p* = 0.0932 (Figure 2B and Appendix A; for this analysis, we excluded the three participants who did not declare gender). Finally, performance was not significantly affected by participants’ institutional roles (F(10,1623) = 0.79, *p* = 0.6400) (Figure 3C; Appendix A).

#### 3.2.6. Confidence

Finally, confidence levels differed according to age (F(5,1628) = 6.91, *p* = 0.0001, η^2^ = 0.02), teaching level (F(9,1624) = 2.82, *p* = 0.0027, η^2^ = 0.02), and time in education (F(7,1626) = 3.59, *p* = 0.0008, η^2^ = 0.02).

In terms of age, the 51–60-year-old group reported the highest overall confidence (4.48), while the 21–30-year-old group reported the lowest confidence (4.20; Appendix A). Post hoc Tukey tests revealed that the 21–30-year-old group reported significantly lower confidence than the 31–40-year-old group (*p* = 0.0001), the 41–50-year-old group (*p* = 0.0001), and the 51–60-year-old group (*p* = 0.0005), respectively.

In terms of teaching level, the multiple levels up to technical school reported the highest confidence (4.47), while the technical school group reported the lowest (4.23; Appendix A). Meanwhile, the only two significant post hoc Tukey comparisons revealed that the pre-school/kindergarten group reported significantly less confidence than the early grammar school (*p* = 0.0135) and multiple levels up to grammar school groups (0.0077), respectively.

Finally, in terms of time in education, the highest confidence was reported by the 31–40 years group (4.5) and the lowest confidence was reported by the less than one year group (4.19; Appendix A). Post hoc Tukey tests revealed that the less than one year group reported significantly lower confidence than the 7–10 years group (*p* = 0.0454), the 11–20 years group (*p* = 0.0173), the 21–30 years group (*p* = 0.0395), and the 31–40 years group (*p* = 0.0010), respectively. Additionally, the 1–3 years group reported significantly less confidence than the 31–40 years group (*p* = 0.0078).

As can be seen in Table 1, increasing scores on individual statements tended to be accompanied by increasing confidence ratings (but see table for a few exceptions).

### 3.3. Analyses by Question Category

Next, we were interested in knowing whether patterns of performance emerged when statements were analyzed by category: (1) brain characteristics, (2) executive and cognitive functions, (3) neurophysiology and learning, (4) emotion and learning, (5) literacy, reading and writing, (6) learning disorders, and (7) learning strategies and methods.

While no significant effects were observed for executive and cognitive functions, emotion and learning, or learning strategies and methods, all other categories showed significant effects.

A multiple regression analysis revealed that statements in the category ‘brain characteristics’ had the greatest influence on performance. For that category (see Table 1), performance differed by teaching levels (F(9,1624) = 2.35, *p* < 0.0125, η^2^ = 0.01), with pre-school/kindergarten teachers performing worst (75.5%) and those teaching multiple levels up to college/university performing best (80.9%). Performance on this category was also better among teachers living in capital cities (78.3%) versus the outskirts (76.9%) (F(1,1632) = 4.51, *p* = 0.0339, d = 0.2).

For neurophysiology and learning, four different variables showed significant effects: (1) teaching level (F(9,1624) = 2.98. *p* = 0.0016, η^2^ = 0.02), with pre-school/kindergarten teachers responding worst (75.9%) and those teaching higher education performing best (86.5%); (2) public versus private (F(2,1631) = 3.17, *p* < 0.0421, d = 0.4), with teachers in private schools performing significantly better than those in public schools (81.9% vs. 78.6%, respectively); (3) capital versus outskirts (F(1,1632) = 4.84, *p* = 0.0280, d = 0.3), with those in capitals performing significantly better than those in the outskirts (81.5% vs. 78.7%, respectively); and (4) region (F(4,1629) = 3.49, *p* = 0.0076, η^2^ = 0.01), with teachers in the South and Southeast performing significantly better than those in the Midwest, North and Northeast.

For literacy, reading and writing, only region was critical (F(4,1629) = 3.71, *p* = 0.0051, η^2^ = 0.01), with teachers in the North responding best (97.1%) and those in the Midwest responding worst (91.9%).

Finally, for learning disorders, two variables were important: (1) teaching level (F(9,1624) = 2.69, *p* = 0.0042, η^2^ = 0.02), with pre-school/kindergarten teachers performing worst (92.9%) and those teaching multiple levels up to college/university performing best (98.5%), and (2) region (F(4,1629) = 2.59, *p* = 0.0255, η^2^ = 0.01), with teachers in the Southeast performing best (96.3%) and those in the Midwest performing worst (91.3%).

Appendix A also shows differences in performance between groups (region, capital vs. outskirts, public vs. private, teaching levels) on 12 of the individual survey statements.

## 4. Discussion

To test neuroscience-related knowledge among educators in Brazil, we created a 28-item survey including general brain-related knowledge as well as common neuromyths, with a special focus on those directly pertaining to education and questions of particular interest to the Brazilian population. A total of 1634 respondents provided anonymous online answers by indicating ‘agree’ or ‘disagree’ to each of the statements, along with their degree of confidence in those answers, and consented to having their responses used in our research study.

When everyone was analyzed together, overall correct responses ranged from 31 to 99%, and confidence levels were generally high (3.48–4.79), differing only by age, teaching level and time in education (see Table 1).

Overall, performance did not differ by age. In contrast to respondents in the United States (laypeople and educators; Macdonald et al., 2017) and those in a previous study from our group conducted with laypeople in Brazil [30], the best scores in the current study were not obtained by the youngest participants. In fact, in the current study, educators with more than 40 years of teaching experience scored best (Figure 3E and Appendix A). This finding is encouraging, as it indicates that educators are able to improve their knowledge over the years.

Representative of the field of education worldwide [22], the vast majority of the educators who responded to our survey were female. Unlike Americans [34] but similar to Greek prospective teachers [22], performance among our educators did not differ by gender. Moreover, performance here was also not influenced by institutional role. All other variables studied influenced performance, as we discuss next.

In terms of region, people in the Southeast performed best, while people in the Northeast and Midwest performed worst (see Figure 2C). Moreover, as a group, individuals from capital cities performed significantly better than those living and teaching in the outskirts of major cities (Figure 2D). These findings reflect the uneven distribution of education and overall access to different resources across and within regions in Brazil [38] and should be studied in other countries to assess whether similar scenarios are observed.

Overall, individuals working in private institutions performed better than those working in public ones (Figure 3B). Interestingly, while educators teaching in municipal and state public schools performed worst and those teaching in private institutions were near the top, the best-performing group were educators in federal public institutions. It should be noted that in Brazil, as in many countries in Latin America, K-12 private schools tend to be better than public schools, but the reverse is seen at the college/university level. Specifically, public universities are often better ranked and more difficult to get into. In the current study, we asked respondents to indicate the level at which they taught as well as the type of institution. In both cases, they also had the option of indicating whether they taught at multiple types of institutions and grade levels. Because these were single answers that did not provide further details, for many respondents, we were not able to identify which grade levels corresponded to private versus public institutions. Therefore, we are not able to identify whether those who responded better and taught at public institutions were in fact teaching mainly in higher education and not lower grades. Future studies could design their questions differently to address this issue specifically.

In terms of teaching level, educators who taught college/university or multiple levels up to college/university performed significantly better than those teaching lower levels (see Figure 3D and Appendix A). In contrast, Dekker et al. (2012) found that knowledge or neuromyth endorsement did not differ between British and Dutch educators who taught primary versus secondary school. In Brazil, teachers of lower grades (kindergarten to primary school) consistently performed below their colleagues teaching higher levels, suggesting quality training for these teachers may not be as readily available. It is likely the case that educators teaching higher levels are required to complete more extensive training and even obtain graduate or specialty degrees, something that is not required of teachers working at the lower stages of education. Importantly, lower overall knowledge and higher endorsement of neuromyths may be particularly harmful among teachers teaching lower grades, as this is the best time to adequately identify and address developmental issues, including disorders that would benefit from early interventions.

### 4.1. Findings by Statement Categories

When we analyzed performance on the statements grouped by category (see Table 1), a few interesting patterns emerged in terms of performance versus confidence. In order of highest to lowest overall scores, the categories ranked as follows: (1) Learning disorders, (2) Literacy, reading and writing, (3) Executive and cognitive functions, (4) Emotion and learning, (5) Learning strategies and methods, (6) Neurophysiology and learning, and (7) Brain characteristics. Interestingly, confidence from highest to lowest ranked as follows: (1) Emotions and learning, (2) Learning strategies and methods, (3) Executive and cognitive functions and Literacy, reading and writing (these two categories were tied for confidence), (4) Neurophysiology and learning, (5) Learning disorders, and finally (6) Brain characteristics. The overall scores reveal that Brazilian educators are better informed about learning disorders and literacy, but learning strategies, neurophysiology and brain characteristics fall behind, which suggests they would benefit from courses or training in these areas of neuroscience. Interestingly, their confidence is lower for the more biology-based themes that they struggle with most (neurophysiology and brain characteristics) but also for learning disorders, which they answered best. Importantly, they seem relatively confident about their knowledge of learning strategies and methods, despite scoring relatively lower on that category. This suggests training and courses designed specifically for educators should focus not only on neuroscience knowledge but also on strategies and methods that educators can specifically incorporate into their teaching. Finally, their confidence on statements regarding emotions and learning was highest, even though these were not their highest scores. Could it be that more technical, science-based questions about brain structure and function simply make educators more insecure (especially if they have had little science-based training or course offerings), while issues that people may consider less difficult, such as emotions, make people more confident in their answers, regardless of knowledge? We are not implying that emotions and the brain are a simple area of study, but perhaps it is a common (albeit erroneous) belief that this may be an easier (or at least more intuitive) topic to master. A future experimental design that could further investigate these nuances would be very interesting.

In terms of these question categories, we also observed significant differences among the variables studied (Appendix A). Specifically, performance on statements about Neurophysiology and learning was influenced by (1) teaching level (with educators in college/university performing consistently better than those teaching lower levels), (2) region (with South/Southeast performing better than all others), (3) capital versus outskirts (with educators from capital cities performing consistently better), and (4) private versus public (with private educators performing significantly better). Moreover, statements about brain characteristics also yielded higher scores among educators in college/university and those living and teaching in capital cities versus the outskirts. These results suggest that more neuroscience-based topics such as neurophysiology and brain characteristics may be more accessible to more ‘privileged’ groups (i.e., working in higher learning and thus potentially having had access to more training and courses; those living in richer regions with access to more resources; and teachers in capital cities and private schools, who also enjoy greater access to resources). Similarly, statements about learning disorders were answered best by educators in college/university and by educators in the Southeast (with educators in the Midwest answering worst). This finding is worrisome as it indicates that knowledge about learning disorders is not as high among educators in lower school levels, a group that is especially critical for identifying and addressing such disorders and potentially facilitating early interventions. Finally, statements about literacy, reading and writing (5) were influenced by region with a surprising result: educators in the North answered best on this category, with those in the Midwest answering worst. Here, a deeper analysis yields a few explanations.

Since 1915, several national and local projects have been implemented to improve literacy in Brazil, some of these led by the Ministry of Education. According to INEP (*Instituto Nacional de Estudos e Pesquisas Educacionais Anísio Teixeira*; National Institute for Educational Research and Studies; http://inep.gov.br/web/guest/inicio, accessed on 15 January 2022), literacy has slowly improved in Brazil thanks to ‘*Brasil Alfabetizado*’, a program implemented by the Ministry of Education in 2003 (http://portal.mec.gov.br/brasil-alfabetizado/apresentacao; accessed on 15 January 2022), with illiteracy falling 0.2% between 2018 and 2019. However, according to a study conducted by PNAD (Pesquisa nacional por amostra de domicílios; national research based on household samples; http://portal.mec.gov.br/programa-mais-educacao/190-secretarias-112877938/setec-1749372213/12521-informacoes-gerais-sobre-a-pnad; accessed on 15 January 2022) of the IBGE (https://www.ibge.gov.br/; accessed on 15 January 2022), literacy rates in the North have actually gone up 91%, presumably as a result of several small initiatives, including PAS (Programa Alfabetização Solidária; https://www.educabrasil.com.br/alfabetizacao-solidaria/; accessed on 15 January 20222) and Tempo de Aprender (Time to learn; https://www.educabrasil.com.br/alfabetizacao-solidaria/; accessed on 15 January 2022), the most complete literacy program in Brazil’s history created by the Ministry of Education especially for public school children in grades K-12 (see also http://alfabetizacao.mec.gov.br/#pna; accessed on 15 January 2022). One critical effort in the region was a collaboration between nearby school districts to implement these initiatives. These data, along with the finding that educators in the North performed best on statements regarding literacy, reading and writing, are extremely encouraging, as they suggest that large-scale and even smaller regional efforts can move education in the right direction.

### 4.2. Brazil versus the World

Our next analysis was to see how Brazilian educators fared relative to educators in Europe, the US and Latin America. In the UK and the Netherlands, Dekker et al. (2012) reported that 49% of educators endorsed neuromyths, while overall accuracy on questions pertaining to general knowledge about the brain was significantly higher, at 70%. Similarly, in our study, Brazilian educators scored worse on classic neuromyths than general questions, but rates here were somewhat higher: 74% on neuromyths and 89% on all other questions. In terms of confidence, Brazilian educators reported 4.15 for neuromyths and 4.43 for all other statements.

For the questions that we specifically selected from previous studies, some interesting patterns emerged. For simplicity, whenever authors reported percent endorsement, we converted that to accuracy scores (i.e., 30% endorsement for a given neuromyth means respondents answered with 70% accuracy on that statement). Comparisons to participants in the US are those reported by Macdonald et al. (2017) [34]; participants in the UK and the Netherlands are those reported by Dekker et al. (2012) [3]; participants in Greece are those reported by Papadatou-Pastou et al. (2017) [22]; and respondents in other Latin American countries are those reported by Gleichgerrcht et al. (2015) [24] (Table 2).

For Question 1 (We only use 10% of our brain’s capacity, correct answer ‘disagree’), only 52.3% of our respondents answered correctly, and the mean level of confidence was 3.95. Interestingly, educators in lower levels (pre-school/kindergarten and grammar school) scored mostly below chance, while those teaching college/university and multiple levels up to college/university scored significantly higher, at 62% and 64%, respectively, which is closer to the scores reported for American educators. Thus, Brazilian educators in private institutions and those teaching higher levels performed as well as US educators on this question, but those in public institutions or teaching lower levels scored significantly lower. In comparison, laypeople who answered this question recently in a study published by our group [30] performed relatively worse (45%), and, quite discouragingly, laypeople in Rio de Janeiro tested 20 years ago were considerably more accurate (68%) [28].

For Question 8 (Teachers should employ teaching methods that stimulate the right side (creative) or the left side (rational) depending on the type of student, correct answer ‘disagree’), overall accuracy among Brazilian educators was 64% and average confidence was 4.08. The scores on this question varied greatly across countries (see Table 2).

For Question 21 (The sleep cycle changes during adolescence; thus, pushing back the start time for morning classes could facilitate learning, correct answer ‘agree’), our educators responded with 62.4% accuracy and 3.95 confidence. A similar statement (Circadian rhythms shift during adolescence causing students to be tired during the first lessons at school) was responded with similar accuracy by others (see Table 2). Although research on shifting circadian clocks over the lifespan is not necessarily recent [39,40,41], studies showing the benefits of more sleep due to later school start times (e.g., better scores, higher IQs, fewer substance abuse and behavioral problems, and even fewer traffic accidents; [42,43,44]) have gained considerable attention in recent years, at least in the United States. Such findings have led to efforts to influence education policies and overall school culture [45]. It is possible that such findings may not have had such an impact outside the US and, thus, this type of issue may not be so well known in other places. Anecdotally, problems in education in Brazil seem to be of a much larger, urgent scale, which may explain why demands for later school start times may not have been brought to the public’s attention. In Brazil, educators teaching in capital cities scored significantly higher (66%) on this question than those in the outskirts (60%), and those in the Southeast scored the highest (65%), followed by the South (63%), Northeast (53%), North (50%) and Midwest (45%).

For Question 24 (Seven years old is considered a critical age for written language acquisition; thus, someone who does not learn to write by this age will most likely never become a good reader, correct answer ‘disagree’), participants scored 90.1% overall with a confidence rate of 4.25 (85%). The other studies used a similar question: There are critical periods in childhood after which certain things can no longer be learned (see Table 2). Such a high score among Brazilian educators is striking and may be partly explained by the national and local projects targeting literacy that have been developed and implemented in recent years and described above. It must also be said that our survey was voluntary, required interest in the area and access to the internet. Despite having collected a large and varied sample that displayed significant differences among groups according to region, teaching level, and type of school, these inherent and inevitable limitations of our study design may have biased our sample to include more well-informed participants in generally more advantageous settings.

For Question 25 (Learning to speak a second language during childhood (bilingualism) can compromise development, correct answer ‘disagree’), our educators responded with 95.7% accuracy and 4.46 confidence. US teachers also had high scores (82%) on a similar question (It is best for children to learn their native language before learning a second language). For the Latin American and European surveys, the question was ‘Children must acquire their native language before a second language is learned. If they do not do so neither language will be fully acquired.’ Scores were somewhat lower for Latin American respondents and significantly lower for European teachers (see Table 2). This finding was surprising, as it is quite common for Europeans to be exposed to more than one language very early in life. The notion that learning a second language too early could be detrimental or is not recommended would seem to be more in line with cultures outside Europe. It is also true that the need to participate more actively in a globalized world has made people in Latin America seek English (or other) language classes, and at increasingly younger ages for those who have the means to acquire them.

Two other statements were significantly influenced by the variables studied. For Question 17 (Dyslexia is a brain condition that is not related to intelligence; thus, an individual can be dyslexic as well as intelligent, correct answer ‘agree’), overall accuracy and confidence levels were high, at 97.1% and 4.56, respectively. However, in terms of teaching levels, pre-school/kindergarten performed significantly worse than most other groups, and once again, educators in the Southeast performed best while those in the Midwest performed worst. As stated above, the first finding is troubling considering dyslexia is often identified by teachers in early education, which is the best time to intervene and help students overcome their difficulties as early as possible.

For Question 20 (Male brains have greater capacity for logical reasoning (science, mathematics), while females are more intuitive (language, arts), correct answer ‘disagree’), overall accuracy was 70% and confidence was 4.02. On this question, educators in capital cities performed significantly better than those living and teaching in the outskirts, and educators in the Southeast performed best, while those in the North performed worst. While overall accuracy on this question may not be as low as others, it still shows that almost 1/3 of educators in Brazil believe in differences in learning abilities or styles between boys and girls or men and women, which is very problematic.

Finally, we analyzed the statement that yielded the lowest score: Children acquire increasingly complex abilities thanks to the process of myelination, which ends after adolescence (19; overall score 31.1%). For this statement, the only variable that influenced performance was teaching level, with college/university and multiple levels up to college/university educators performing best, both of which differed significantly from pre-school/kindergarten teachers.

In sum, our survey revealed that overall, Brazilian educators’ knowledge of the brain is relatively high (compared with other studies conducted in other Latin American countries, Europe and the United States). Similar to teachers in other countries, Brazilian educators endorsed several common neuromyths, as accuracy (and confidence) on these items was lower than that of general knowledge items. Performance was lowest on questions pertaining to brain characteristics, neurophysiology and function. There is a strong belief in classic neuromyths, including the use of only 10% of the brain, and more ‘dangerous’ myths, such as differences in learning abilities between the sexes and innate hemispheric specializations (left- vs. right-brained individuals).

Overall, performance was significantly lower among educators living in the North, Northeast and Midwest than those living and teaching in the South and Southeast, reflecting longstanding inequalities across regions regarding education and general access to resources. However, statements regarding literacy, reading and writing were answered best by educators in the North, which may reflect the recent implementation of national and local projects targeting these less-favored regions. Performance was also lower among educators living in the outskirts versus the capitals of major cities, and among educators in public versus private institutions. Critically, overall knowledge was relatively lower among educators teaching lower levels (pre-school, kindergarten, and grammar school) relative to those teaching college/university. This finding is particularly worrisome, considering the early years are key for developing basic skills and for identifying difficulties that may benefit from early intervention. Educators at this level who have poor knowledge of brain function and learning disorders combined with preconceived misguided notions of individuals’ innate abilities or capacity to learn can have long-lasting negative effects on the education and future of their students.

### 4.3. What to Do with These Findings?

How can this scenario be improved? What kinds of information or courses should educators seek to improve their knowledge? In a simple Google search we conducted in November of 2020, we found more than 400 course offerings in Neuroscience or Neuroeducation in Brazil, most of them low cost or even free. Fewer than 10% of these courses are associated with an accredited higher learning institution, and less than 3% of graduate-level extension courses are offered by universities that are well placed in the Ministry of Education’s (MEC) most recent general course index [46]. In sum, there seems to be a growing supply of courses in this field, but it is difficult to assess the quality of these new courses, especially at this early stage.

Several authors have discussed ideas regarding how neuroscience courses should be designed for educators. Most agree that content should be less technical and more adapted to themes relevant to the field of education. Additionally, education-related neuromyths should be tackled head-on to be as effective as possible. Finally, courses should emphasize the critical thinking skills necessary to critique information and to read and understand primary sources of information [3,12,14,22,47,48,49,50]. Without these skills to make them independent thinkers and seekers of knowledge, educators are vulnerable to fake or misrepresented knowledge as well as products associated with such pseudoscience. In Brazil, as in other developing countries, we have the added difficulty of poor English language skills and generally lower resources (institutional or personal) that educators can invest in their own training and education [24]. However, as the literature shows, educators living and working in countries where these are not pressing problems still fall prey to misinformation. Scientists all over the world should come together to think of solutions to this problem to discover which approaches would work best in each region.

In the current study, we gathered information about neuroscience knowledge and neuromyth endorsement among Brazilian educators in all five regions of the country, different teaching levels and types of school. Although our reach was wide, the survey was voluntary, and our respondents needed to have a working internet connection, as well a basic interest in the role that neuroscience plays in education. Thus, our design does not give us a picture of the situation in rural, poorer regions without internet access or in areas where limited training or course offerings for educators may fail to incite this kind of interest. Nonetheless, we were able to identify that certain national and local projects may have contributed to improve the scenario in the North, which is usually disadvantaged, at least in terms of literacy, reading and writing, suggesting similar efforts should target the Midwest and the Northeast. Training in neuroscience knowledge and learning disorders, but perhaps more critically in knowledge directly applied to teaching (and dispelling neuromyths), should also primarily target lower school teachers, as well as those working in public schools and in the outskirts of major cities. We hope these findings guide us and others towards initiatives that seek to lessen these longstanding inequalities in an effort to democratize education as much as possible, even within the limitations encountered in any nation, regardless of level of development.

## Figures and Tables

**Figure 1 brainsci-12-00734-f001:**
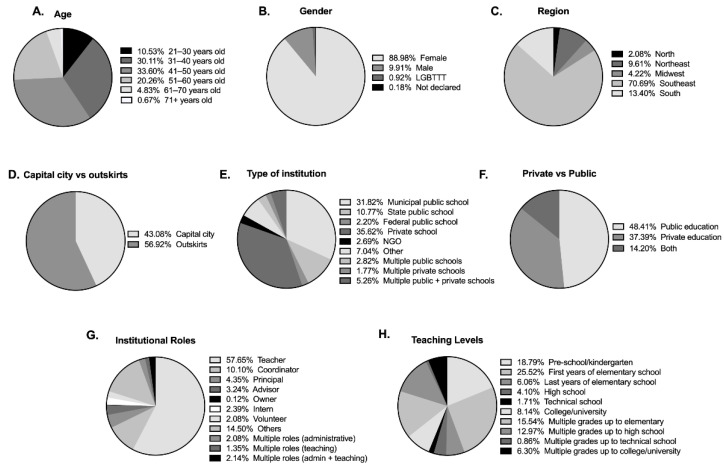
Distribution of respondents (*n* = 1634) according to age group. Mult.: Multiple. LGBTTT: lesbian, gay, bisexual, transvestite, transexual and transgender. NGO: non-governmental organization.

**Figure 2 brainsci-12-00734-f002:**
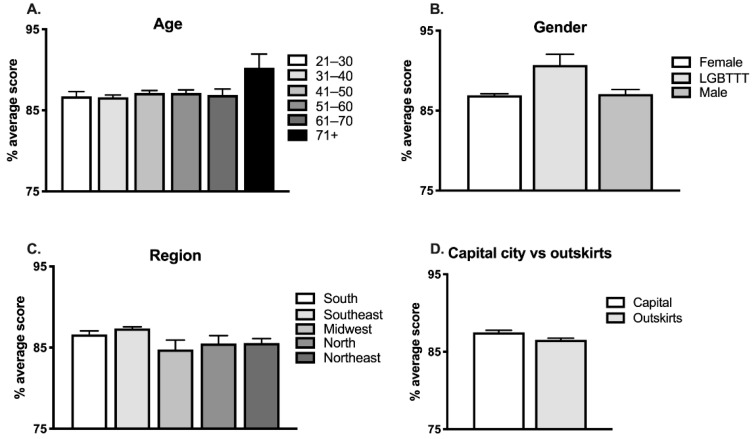
Effects of age, gender and geographical region on performance.

**Figure 3 brainsci-12-00734-f003:**
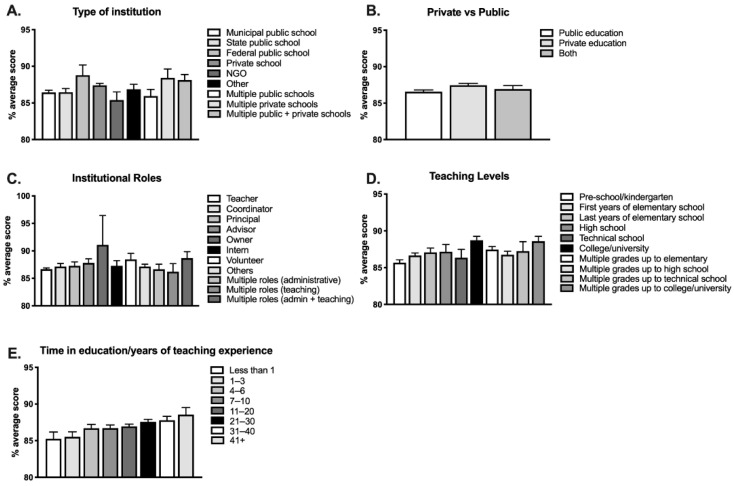
Institutional variables affecting respondents’ performance. Mult.: Multiple.

**Table 1 brainsci-12-00734-t001:** Survey, responses, and confidence levels.

Assertions by Category	CA	Correct (%)	Incorrect (%)	Mean Confidence 1–5
1. Brain characteristics				
Children acquire increasingly complex abilities thanks to the process of myelination, which ends after adolescence (19).	F	31.1	68.9	3.48
We only use 10% of our brain’s capacity (1).	F	52.3	47.7	3.95
Male brains have greater capacity for logical reasoning (science, mathematics), while females are more intuitive (language, arts) (20).	F	70.0	30.0	4.02
Neuroplasticity (the brain’s capacity to change, adapt and learn new things) ends after adolescence (27).	F	93.8	6.2	4.48
Learning maturation (cerebral maturity) depends exclusively on genetics (6).	F	96.2	3.8	4.31
One must keep the brain active in order not to lose (and continue creating) connections between neurons (22).	T	97.9	2.1	4.63
Although neuroplasticity declines with age, it is possible to learn throughout one’s entire life (5).	T	99.1	0.9	4.79
2. Executive and cognitive functions				
Mastering self-control before reaching adulthood is associated with greater prosperity and health throughout life (18).	T	86.4	13.6	4.18
There are no benefits to starting self-control training before adolescence (26).	F	92.8	7.2	4.36
Self-control training in students is exclusively the responsibility of parents, not teachers (7).	F	93.5	6.5	4.43
Learning is more efficient when we recruit different cognitive functions, including memory, attention and the five senses (14).	T	97.9	2.1	4.7
3. Neurophysiology and learning				
The sleep cycle changes during adolescence; thus, pushing back the start time for morning classes could facilitate learning (21).	T	62.4	37.6	3.95
A good night’s sleep, healthy food and regular physical exercise promote learning (3).	T	97.4	2.6	4.78
4. Emotion and learning				
A student’s sadness interferes directly with his/her feelings and not with learning performance (15).	F	76.0	24.0	4.54
When something is learned with emotion, it is easily remembered (9).	T	97.6	2.4	4.74
5. Literacy, reading and writing				
Seven years old is considered a critical age for written language acquisition; thus, someone who does not learn to write by this age will most likely never become a good reader (24).	F	90.1	9.9	4.25
When teaching reading and writing, it is relevant to know that letters do not only represent sounds (alphabetical principle), but also phonemes (smaller sound units of a language) (12).	T	92.4	7.5	4.27
Learning to speak a second language during childhood (bilingualism) can compromise development (25).	F	95.7	4.3	4.46
Left-handed students are not as capable of learning to read as right-handed students (11).	F	96.6	3.4	4.39
During the literacy process, some children may benefit from a combination of teaching methods (10).	T	97.6	2.5	4.71
6. Learning disorders				
Medications are the only proven strategy to treat attention deficit hyperactivity disorder (ADHD) (4).	F	94.3	5.7	4.16
Dyslexia is a brain condition that is not related to intelligence; thus, an individual can be dyslexic as well as intelligent (17).	T	97.1	2.9	4.56
7. Learning strategies and methods				
Teachers should employ teaching methods that stimulate the right side (creative) or the left side (rational) depending on the type of student (8).	F	64.0	36.0	4.08
Neuroimaging techniques such as magnetic resonance imaging (MRI) can reveal a student’s intelligence and capacity to learn (16).	F	68.0	32.0	3.75
Teachers enhance students’ learning by asking questions and not just presenting answers (content) (2).	T	94.3	5.8	4.66
Frequent breaks during classes, as well as alternating between theorical and practical activities, is considered a good teaching strategy that facilitates learning (23).	T	98.4	1.6	4.69
Mindfulness, breathing and meditation techniques can contribute to learning (28).	T	96.6	3.4	4.65
It is important to stimulate both creativity and rational thinking for students to develop fully (13).	T	98.8	1.2	4.79

Table 1 legend: Questions were categorized according to seven categories of knowledge (see text). (number): indicates the order in which each question was presented during the actual survey. All participants viewed and answered all 28 statements in the same order. The table also shows the percentage of correct and incorrect responses, as well as the mean confidence participants reported in their answer (on a scale of 1–5 and as a percentage). In this table, we listed the statements within each statement category in increasing order of mean accuracy across respondents. CA: correct answer; F: False; T: True; %: percent.

**Table 2 brainsci-12-00734-t002:** Brazil versus other countries.

Question	Brazil	US	UK	The Netherlands	Greece	Peru	Argentina	Chile	Other (Latin America)
1. 10%	52	67	48	46	N/A	33	44	59	40
8. right vs. left	64	51	91	86	8	25	42	19	27
21. circadian rhythms in adolescence	62	83	70 *	70 *	35	69	51	55	62
24. 7 years old and writing	90	81	33	52	31	33	29	26	34
25. bilingualism compromises development	96	82	7	36	36	50	84	80	97

Table 2 legend: Comparison of scores in Brazil versus other countries studied on selected questions. Scores listed as percent correct. Other (Latin America) refers to average scores obtained by respondents in Mexico, Nicaragua, Colombia and Uruguay in Gleichgerrcht et al., 2015. Question 1: We only use 10% of our brain’s capacity, correct answer ‘disagree’; Question 8: Teachers should employ teaching methods that stimulate the right side (creative) or the left side (rational) depending on the type of student, correct answer ‘disagree’; Question 21: The sleep cycle changes during adolescence; thus, pushing back the start time for morning classes could facilitate learning, correct answer ‘agree’; Question 24: Seven years old is considered a critical age for written language acquisition; thus, someone who does not learn to write by this age will most likely never become a good reader, correct answer ‘disagree’; Question 25: Learning to speak a second language during childhood (bilingualism) can compromise development, correct answer ‘disagree’. * = Dekker et al. (2012) did not report the score on this particular item but reported 70% accuracy for all items falling into the category of general knowledge about the brain.

## Data Availability

Raw data for this study have been made publicly available at https://doi.org/10.17026/dans-xq3-s2t4 (accessed on 17 December 2021).

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
