# Peer review of "Neuroscience Knowledge and Endorsement of Neuromyths among Educators: What Is the Scenario in Brazil?"

_brainsci, 2022, doi:10.3390/brainsci12060734_

Round 1
Reviewer 1 Report
Simoes et al report on an online survey taken by Brazilian teachers in early 2020 regarding their knowledge of neuroscience that included some statements known to be neuromyths. The other statements appear to be cobbled together from previous work by these authors. The lack of reliability and validity data lower enthusiasm for the instrument used. Survey results were examined for overall correctness, which was generally high, and disaggregated by teachers’ age, gender, region of the country, capital city vs otherwise, and multiple type of school categories. The data and statistical methodologies appear solid. In examining the disaggregated results, even the differences that were reported as significant are at best equal to the answer to a single question and have low effect sizes. Therefore, it is difficult to be enthusiastic about the interpretations provided. Compared to other published results on neuromyths or neuroscience literacy in other countries, the current results appear high. The paper can be improved by addressing the following issues.
How was the survey advertised so that educators were captured as the intended audience?
What are the reliability and validity of the instrument created for this study?
Line 198-9: The following statement regarding data in Table 1 appears to be at odds with the data reported: “Scores were converted to continuous variables (percent correct, 0-100% across all 30 items) for each participant.” A single participant rates a specific item only once, so a percent correct cannot be calculated from that one response.
Lines 197 & 199: Are there 28 statements or 30? If there were 30 at one time, why were 2 discarded?
For performance, the effects of region, type of institution, teaching level, and time in education had very small effect sizes, η2=0.01 or 0.02, despite statistical significance. Of what practical significance is this information? Similarly, for the confidence measure, what is the practical significance of at most a 0.4 change in a Likert value and of their low effect sizes?
What is the effect size of the comparison of the capital cities to elsewhere, public to private schools?
The range of % differences in most of the categories are less than 3-4%. For 28 questions, a 1 question difference is 3.5%. Is variation of this magnitude worth worrying about?
Effect sizes continue to be small when the data are disaggregated by question type. With only small effect sizes, why should the reader care? Does a 1 question average difference really reveal a major lack of teacher education in one category or another?
How were the categories of questions determined? Was a factor analysis done? Why not?
Line 357. Is the study under review, the Front. Hu. Nsci 2022 article or a different one?
Line 359. Fig. 3C is breakdown by institutional roles, not years in teaching. How would one assess years of teaching in a lay population? What comparison is being made here?
While the # are not provided, the comparison for capital vs outskirts appear to vary by at most 1%. How important is that difference?
Line 374. Which figure is being referred to, 3E or 3A?
Line 390. Table S8 does not contain information on teaching levels.
All the figure and table numbers should be double checked throughout the manuscript.
Line 488, the 10% myth is not question 1 in Table 1. It appears first in table S13.
Line 506, ‘teachers should employ…’ is not the 8th question in either table. Be consistent.
Similarly, in lines 517, 539, 555, 572, & 580 the written questions are not ordered as stated.
In the comparison of the Brazilian data to the previously published data from other authors, present the numbers as a table and condense the interpretation paragraphs.
To provide the reader with ease in understanding the comparison made about data in table S13, a column should be added so that each rows’ ‘category’ is stated. Only a subset of the 28 questions from Table 1 appear in Table 13. These should be presented in the same order as table 1.
What do the different levels of self-ratings of confidence in teachers’ neuroscience knowledge really mean? Most of these numbers are reasonably high. Is that justified? Is there a correlation between actual knowledge and confidence?
Bibliographic references do not contain all necessary information. Many are missing the name of the publication, volume and page numbers.
Surprisingly, the original public knowledge of neuroscience paper by Herculano-Houzel, done in Brazil, was not referenced! How does the current data compare with this 2002 paper?
Author Response
Please see the point-by-point responses bellow. Also, we uploaded the track changes manuscript (see the attachment)
Reviewer 1:
Comment:
Simoes et al report on an online survey taken by Brazilian teachers in early 2020 regarding their knowledge of neuroscience that included some statements known to be neuromyths. The other statements appear to be cobbled together from previous work by these authors. The lack of reliability and validity data lower enthusiasm for the instrument used. Survey results were examined for overall correctness, which was generally high, and disaggregated by teachers’ age, gender, region of the country, capital city vs otherwise, and multiple type of school categories. The data and statistical methodologies appear solid. In examining the disaggregated results, even the differences that were reported as significant are at best equal to the answer to a single question and have low effect sizes. Therefore, it is difficult to be enthusiastic about the interpretations provided. Compared to other published results on neuromyths or neuroscience literacy in other countries, the current results appear high. The paper can be improved by addressing the following issues.
How was the survey advertised so that educators were captured as the intended audience?
Response:
In Supplementary Table 1, we list all the education events, journals and social media pages where we advertised the survey. As you can see below, these were education conferences held all over the country where some of the co-authors presented invited lectures. Next to each event, we list the region where the event took place, as well as the number of people reached per event. We also list the Facebook pages of different national and regional educators’ groups where we were allowed to post the survey, along with the number of followers of each of those pages.
S1 Table. Distribution of the survey
|
By email |
||
|
Event |
Region (state) |
Number of people reached |
|
South region Symposium (Lecture) |
SOUTH (RS) |
68 |
|
Educators meeting in Manaus (Lecture) |
NORTH (AM) |
90 |
|
Educators meeting in Fortaleza (Lecture) |
NORTHEAST (CE) |
44 |
|
Pedagogical meeting in Recife (Lecture) |
NORTHEAST (PE) |
785 |
|
Pedagogical meeting in Rio de Janeiro (Lecture) |
SOUTHEAST (RJ) |
734 |
|
Pedagogical meeting in Londrina (Lecture) |
SOUTH (PR) |
240 |
|
Pedagogical meeting in São Paulo (Lecture) |
SOUTHEAST (SP) |
851 |
|
Educators meeting in Palmas (Lecture) |
NORTH (TO) |
101 |
|
25ª Municipal day for teaching and learning (Lecture) |
SOUTH (RS) |
102 |
|
Meeting of private school teachers |
SOUTHEAST (SP) |
48 |
|
Education meeting in Fortaleza (Lecture) |
NORTHEAST (CE) |
42 |
|
Education meeting in Recife (Lecture) |
NORTHEAST (PE) |
46 |
|
Education meeting in Porto Alegre (Lecture) |
SOUTH (RS) |
45 |
|
Education meeting in São Paulo (Lecture) |
SOUTHEAST (SP) |
118 |
|
Via social media (FACEBOOK) |
||
|
Groups |
Region (state) |
Number of people reached |
|
Roraima professors’ group |
SOUTHEAST (SP) |
3,000 followers |
|
Pernambuco professors’ group |
NORTHEAST (PE) |
28,800 followers |
|
Tocantins professors’ group |
NORTH (TO) |
698 followers |
|
Alagoas state professors’ group |
NORTHEAST (AL) |
4,000 followers |
|
Elementary school professors’ group |
BRAZIL |
288,700 followers |
|
Professors syndicate group |
NORTH & NORTHEAST |
8,055 followers |
|
Union of educational establishments in the state of São Paulo (Sieeesp) |
SOUTHEAST (SP) |
3,585 followers |
|
By Journals |
||
|
Brazilian Association of Psychopedagogy |
BRAZIL |
|
|
Federal Council of Education |
BRAZIL |
|
Comment:
What are the reliability and validity of the instrument created for this study?
Response:
The current survey was compiled from different sources, each with its own established reliability/validity, as follows: 1) the classic neuromyths tested in several previous studies around the world which have been reliably tested and retested in different languages; 2) our previous survey tested on laypeople (Arévalo et al., 2022), in which we used Google AdWords to identify the most common terms (and associated questions) searched by Brazilians in Portuguese, and 3) the pilot studies conducted with educators by co-author AF in education conferences all over Brazil between 2015-2017 (see Introduction). The main objective of this study was to create a new survey that not only tested concepts previously tested in different parts of the world, but also included specific doubts and interests expressed by Brazilian laypeople and educators. Thus, as a new survey, while its contents have been tested elsewhere, the reliability and validity of its current form will need to be further tested in future applications (ideally with the same or similar groups of educators in Brazil).
Comment:
Line 198-9: The following statement regarding data in Table 1 appears to be at odds with the data reported: “Scores were converted to continuous variables (percent correct, 0-100% across all 30 items) for each participant.” A single participant rates a specific item only once, so a percent correct cannot be calculated from that one response.
Response:
We apologize if this was not clear enough in the manuscript. Percent correct was calculated 1) for each participant and across items -- all 28 items responded by each participant, for total overall percent correct; and 2) for each item and across participants – all responses to each of the 28 items for all 1634 participants. We have revised the manuscript accordingly.
Comment:
Lines 197 & 199: Are there 28 statements or 30? If there were 30 at one time, why were 2 discarded?
Response:
There were 28 statements. We corrected the typo in the revised manuscript.
Comment:
For performance, the effects of region, type of institution, teaching level, and time in education had very small effect sizes, η2=0.01 or 0.02, despite statistical significance. Of what practical significance is this information? Similarly, for the confidence measure, what is the practical significance of at most a 0.4 change in a Likert value and of their low effect sizes?
Response:
You are right in noticing that the effect sizes were small, despite significance in the original analyses (ANOVAs and t-tests). While we were able to test a relatively large group of participants that were well-distributed among regions and other variables tested, we used a snowball sampling method, which does not allow us to equally distribute participants across categories or even ensure our sample is representative of the larger group of Brazilian educators. That being said, we still believe our data make an important contribution to this line of inquiry and set the stage for bigger exploratory studies. Overall, the comparisons with the largest effects were region, followed by time in education, then capital versus outskirts and teaching level. We chose to tone down the interpretation of some of these results in the revised manuscript to paint a clear picture of this contribution.
Comment:
What is the effect size of the comparison of the capital cities to elsewhere, public to private schools?
Response:
The effect size of the comparison between capital cities and outskirts was Cohen’s d=0.15 (please see line 249 of the revised manuscript), while that of private versus public school educators was d=0.35 (please see line 269).
Comment:
The range of % differences in most of the categories are less than 3-4%. For 28 questions, a 1 question difference is 3.5%. Is variation of this magnitude worth worrying about?
Effect sizes continue to be small when the data are disaggregated by question type. With only small effect sizes, why should the reader care? Does a 1 question average difference really reveal a major lack of teacher education in one category or another?
Response:
We agree with the reviewer that the effect sizes are small and thus should be interpreted with caution. We do believe the results reveal trends that are interesting to pursue in future studies and thus we have chosen to reframe our text to better express this interpretation.
Comment:
How were the categories of questions determined?
Response:
The questions were chosen based on several criteria, as mentioned in the methods (previous studies on neuromyths, themes covered in neuroscience and education courses and conferences, questions asked by students and educators, Google AdWords results of terms most commonly searched for in Brazil and in Portuguese, etc.). Once we chose the ‘best’ questions, we divided them into the most logical categories (1. brain characteristics, 2. executive and cognitive functions, 3. neurophysiology and learning, 4. emotion and learning, 5. literacy, reading and writing, 6. learning disorders, and 7. learning strategies and methods).
Comment:
Was a factor analysis done? Why not?
Response:
We did run multiple regression analyses, both for the original variables studied (age, gender, institutional roles, region, teaching level, type of institution, years of teaching, capital vs. outskirts and public vs. private) and for the question categories (the 7 listed in the previous answer). We have added that information to the text, prior to reporting the ANOVAs.
Comment:
Line 357. Is the study under review, the Front. Hu. Nsci 2022 article or a different one?
Response:
Yes, we have corrected that in the revised text.
Comment:
Line 359. Fig. 3C is breakdown by institutional roles, not years in teaching.
Response:
You are correct. Thank you for noticing this error and the ones in the following comments. We have corrected these throughout the text.
Comment:
How would one assess years of teaching in a lay population? What comparison is being made here?
Response:
We did not survey a lay population. We actively recruited members of the education community. Thus, years of teaching was assessed by having them indicate the number of years they have worked in education.
Comment:
While the # are not provided, the comparison for capital vs outskirts appear to vary by at most 1%. How important is that difference?
Response:
As mentioned in our response to previous comments, we agree with the reviewer that some effect sizes are small and thus should be interpreted with caution. We do believe the results reveal trends that are interesting to pursue in future studies and thus we have chosen to reframe our text to better express this interpretation.
Comment:
Line 374. Which figure is being referred to, 3E or 3A?
Response:
That is 3B. We have corrected the text accordingly.
Comment:
Line 390. Table S8 does not contain information on teaching levels. All the figure and table numbers should be double checked throughout the manuscript.
Response:
You are correct. It is Table S5. We have corrected the text accordingly and have double-checked all other figure and table numbers.
Comment:
Line 488, the 10% myth is not question 1 in Table 1. It appears first in table S13.
Line 506, ‘teachers should employ…’ is not the 8th question in either table. Be consistent.
Similarly, in lines 517, 539, 555, 572, & 580 the written questions are not ordered as stated.
Response:
In Table 1, we organized the statements by category and listed the statements within each category in order of increasing mean accuracy. Therefore, the table does not present the statements in order of presentation; order of presentation appears in parentheses after each statement. So, the 10% myth question was in fact the first statement seen by all participants. The table legend describes this, and we added some information that we hope makes this easier to understand. We chose to present the statements this way (instead of listing them in the actual order of presentation) to facilitate understanding of the different question categories and the scores associated with those.
Comment:
In the comparison of the Brazilian data to the previously published data from other authors, present the numbers as a table and condense the interpretation paragraphs.
Response:
We have created a new table (Table 2) that summarizes the comparison of Brazilian data to that of other countries.
Comment:
To provide the reader with ease in understanding the comparison made about data in table S13, a column should be added so that each rows’ ‘category’ is stated. Only a subset of the 28 questions from Table 1 appear in Table 13. These should be presented in the same order as table 1.
Response:
We have added a column to Table S13 entitled ‘category’, as well as the order of presentation of each question in parentheses, as we did in Table 1. The questions listed in Table S13 are questions for which significant differences in performance were observed for any of the variables tested. We did not include the 16 questions for which there were no significant differences. We hope the changes make the table easier to read.
Comment:
What do the different levels of self-ratings of confidence in teachers’ neuroscience knowledge really mean? Most of these numbers are reasonably high. Is that justified? Is there a correlation between actual knowledge and confidence?
Response:
The confidence score was an additional piece of information we collected to see whether any interesting patterns would emerge. As we discuss throughout the paper, confidence was relatively high throughout. However, more interesting is the observation that confidence seemed to be higher for more ‘accessible’ themes like emotions, and lower for more ‘brain’ and ‘neurophysiology’-related themes, even when scores did not match these confidence levels. While we are not implying that the study of emotions is simple, we do believe laypeople as well as educators may interpret it to be. Emotions may be interpreted as being more intuitive and perhaps not as intimidating as more biology-based concepts. Thus, educators may not feel confident about topics that may seem more difficult to them or that may not be covered as much in courses designed for them. In addition, overall, neuromyths yielded lower scores (and lower confidence) than general knowledge statements, suggesting this is an area courses for educators should focus on. Thus, courses designed to improve educators’ knowledge of the brain and how neuroscience may be used to improve education should take confidence into consideration.
Comment:
Bibliographic references do not contain all necessary information. Many are missing the name of the publication, volume and page numbers.
Response:
References that seem incomplete are actually not articles published in scientific journals; they refer to websites or other sources that report on general media surveys or government websites. In those cases, we have provided the actual URLs readers can access for more information.
Comment:
Surprisingly, the original public knowledge of neuroscience paper by Herculano-Houzel, done in Brazil, was not referenced! How does the current data compare with this 2002 paper?
Response:
We have added this reference to the introduction as well as the discussion.
Reviewer 2 Report
This manuscript details a descriptive study assessing Brazilian educators’ knowledge of neuroscience, by asking them to rate 28 statements framed as being about neuroscience, many of which are considered common neuromyths, or incorrect beliefs about neuroscience and/or education. The sample was large (over 1600 participants) and everyone answered the same survey. The accuracy in assessing each statement ranged from 31% to 99%, and it appears the overall average was above 85% correct, so the educators in the sample had a pretty high level of knowledge. (It might be useful to report the overall correct percentage instead of just breaking it down by different categories.)
Some comments:
There were many significant findings, but that is perhaps not that surprising with such a large sample; even small differences in magnitude are likely to be significant.
In looking at the chosen statements participants were asked about, I am not sure I would classify them all as neuroscience-related. Some are more directly about cognition and/or education. Yes, they are built to a degree on neuroscience foundations, but learning neuroscience on its own would not necessarily help in learning, for example that “During the literacy process, some children may benefit from a combination of teaching methods” or that “Teachers enhance students’ learning by asking questions and not just presenting answers (content).” You might want to think about reframing somewhat. This point is related to my point above that the accuracy was very high on many statements, suggesting they are not in fact common neuromyths, at least among this population.
I advise being cautious with causal language when talking about non-causal participant variables. For example, saying “region had a significant effect on participants’ scores” is probably not quite right, in that there is not random assignment, so we typically talk about associations.
In section 3.2.6, review the scale of confidence measures. I was also interested in links between confidence and accuracy, as a rough assessment of participants’ metacognitive awareness of their neuroscience understanding. Looking at the confidence levels reported for each question in Table 1, it appears there is a relationship between accuracy and confidence, albeit an imperfect one.
The Discussion is very long and spends a lot of time repeating results already reported in the Results section and mentioning some that were not included there, such as comparisons between Brazilian educators and those from other countries. I am torn about the best way to present this, as obviously the non-Brazilian data was not newly-collected for this study, yet it also seems a little odd to compare it in such detail only in the Discussion. Overall, though, I think some of the details could safely be cut (or at least moved to the supplemental section).
I wonder if it might also help to organize the Discussion with some subheadings, to make it easier to follow instead of six pages of solid text with no clear organization. Also, trimming some text that repeats points from the paper would streamline the arguments. Focusing more of the Discussion on potential explanations for the findings (and thus avenues for future research) would strengthen the argument in the paper instead of going through so much detail in the responses to specific questions.
Overall, I think this is an interesting descriptive study that empirically documents some evidence of understanding about education and neuroscience research in an under-studied population. I think the paper could be improved with some revisions as noted above.
Author Response
Please see the point-by-point responses bellow. Also, we uploaded the track changes manuscript (see the attachment)
Reviewer 2:
This manuscript details a descriptive study assessing Brazilian educators’ knowledge of neuroscience, by asking them to rate 28 statements framed as being about neuroscience, many of which are considered common neuromyths, or incorrect beliefs about neuroscience and/or education. The sample was large (over 1600 participants) and everyone answered the same survey. The accuracy in assessing each statement ranged from 31% to 99%, and it appears the overall average was above 85% correct, so the educators in the sample had a pretty high level of knowledge. (It might be useful to report the overall correct percentage instead of just breaking it down by different categories.)
Some comments:
Comment:
There were many significant findings, but that is perhaps not that surprising with such a large sample; even small differences in magnitude are likely to be significant.
In looking at the chosen statements participants were asked about, I am not sure I would classify them all as neuroscience-related. Some are more directly about cognition and/or education. Yes, they are built to a degree on neuroscience foundations, but learning neuroscience on its own would not necessarily help in learning, for example that “During the literacy process, some children may benefit from a combination of teaching methods” or that “Teachers enhance students’ learning by asking questions and not just presenting answers (content).” You might want to think about reframing somewhat. This point is related to my point above that the accuracy was very high on many statements, suggesting they are not in fact common neuromyths, at least among this population.
Response:
We thank the reviewer for this comment. Indeed, because our paper targeted educators, we were interested in testing both common neuromyths as well as more general themes relevant to education (with varying degrees of ‘brain’ involved). In fact, the way we categorized the questions illustrates these distinctions. Regarding how much neuroscience itself can influence education (or how much neuroscience is needed to train good educators) remains a relatively controversial discussion that we do not intend to get into in this manuscript. Our main goal was to survey knowledge about themes at the interface of neuroscience and education and identify patters that could be addressed to better serve educators interested in further knowledge.
Comment:
I advise being cautious with causal language when talking about non-causal participant variables. For example, saying “region had a significant effect on participants’ scores” is probably not quite right, in that there is not random assignment, so we typically talk about associations.
Response:
Thank you for this comment. We agree with the reviewer and have revised the text accordingly.
In section 3.2.6, review the scale of confidence measures. I was also interested in links between confidence and accuracy, as a rough assessment of participants’ metacognitive awareness of their neuroscience understanding. Looking at the confidence levels reported for each question in Table 1, it appears there is a relationship between accuracy and confidence, albeit an imperfect one.
Response:
We have added this information to section 3.2.6.
Comment:
The Discussion is very long and spends a lot of time repeating results already reported in the Results section and mentioning some that were not included there, such as comparisons between Brazilian educators and those from other countries. I am torn about the best way to present this, as obviously the non-Brazilian data was not newly-collected for this study, yet it also seems a little odd to compare it in such detail only in the Discussion. Overall, though, I think some of the details could safely be cut (or at least moved to the supplemental section).
Response:
Thank you for this comment. We decided to add a table (Table 2) to summarize much of the information presented in the Discussion. We hope this reduces the text a bit and helps streamline the reading.
Comment:
I wonder if it might also help to organize the Discussion with some subheadings, to make it easier to follow instead of six pages of solid text with no clear organization. Also, trimming some text that repeats points from the paper would streamline the arguments. Focusing more of the Discussion on potential explanations for the findings (and thus avenues for future research) would strengthen the argument in the paper instead of going through so much detail in the responses to specific questions.
Response:
Thank you for this useful suggestion. We have cut some text and also added subheadings to better organize the Discussion. We hope these changes have improved the manuscript.
Comment:
Overall, I think this is an interesting descriptive study that empirically documents some evidence of understanding about education and neuroscience research in an under-studied population. I think the paper could be improved with some revisions as noted above.
Response:
We thank you for your helpful comments and hope our revisions have improved our manuscript.